# Mechanism Design Meets Large Language Models: Foundations and Frontiers

A& B

**Abstract**

Large language models (LLMs) increasingly operate in interactive and strategic settings such as negotiation, preference elicitation, evaluation, alignment, and multi-agent coordination. These settings are inherently incentive-driven, yet most existing approaches rely on heuristic designs with limited guarantees against manipulation or misalignment. This survey develops a unified, bidirectional view of the relationship between mechanism design and LLMs. We discuss (i) LLMs for mechanism design, where language models are used as strategic agent proxies, simulators, and natural-language interfaces for economic mechanisms, and (ii) mechanism design for LLMs, where incentive-aware principles are applied to LLM evaluation, alignment, and training. We conclude by identifying challenges in incentive-compatible evaluation, human-LLM interaction protocols, and emerging markets for LLM services.

**Keywords:** Large language models, Mechanism design, Incentive compatibility, Multi-agent systems

**Mathematics Subject Classification (2020):** 62XXX

## 1 Introduction

LLMs are increasingly deployed in interactive and strategic settings, including negotiation, preference elicitation, evaluation, alignment, and multi-agent coordination. Modern LLMs have evolved from passive text generators into agentic systems capable of planning, multi-step reasoning, negotiation, and interaction in complex environments (et al., 2024; Guo et al., 2024b; Wang et al., 2023; Acharya et al., 2025). In these settings, LLMs do not operate in isolation; they respond to feedback, adapt to incentives, and interact with humans (Chen et al., 2025d; Frankel, 2014). Consequently, the behavior of LLM-based systems is shaped not only by model architectures and training data, but also by the rules governing interaction, feedback, and information flow (Pan et al., 2024; Ouyang et al., 2022). Despite this shift toward interactive deployment, most current approaches to LLM-based agents and alignment rely on heuristic design choices, including prompt engineering, reward shaping, and ad hoc evaluation protocols, with few formal guarantees against manipulation, strategic reporting, or incentive misalignment (Naik et al., 2025; Anwar et al., 2024). While these methods can perform well in static or single-agent settings, they often break down in interactive and decision-making pipelines, where LLMs adapt to feedback and exploit weaknesses in the design (Lorè and Heydari, 2024; Hao and Duan, 2025; Buening et al., 2025). A principled framework that explicitly accounts for incentives and strategic interaction is therefore essential.

Mechanism design, a branch of economics and game theory, studies how to design rules of interaction so that self-interested agents generate desirable outcomes (Börgers, 2015). It provides algorithmic tools for handling incentives, information asymmetry, and equilibrium behavior, and has played a central role in the design of auctions (Myerson, 1981), matching markets (Roth and Sotomayor, 1992), resource allocation mechanisms (Hurwicz, 1973), and broader market design applications (Myerson, 2013). While mechanism design has long guided the analysis of strategic human behavior, its relevance to modern language-based AI systems has only recently become apparent and remains fragmented across disciplines.

This convergence makes the connection between mechanism design and LLMs both natural and increasingly necessary. On one hand, LLM alignment and reinforcement learning fundamentally rely on the design of rewards and feedback, which is itself a mechanism design problem (Ji et al., 2025; Munos et al., 2024). When multiple LLMs interact, they form multi-agent systems that may exhibit strategic behavior, coordination failures, or manipulation (Li et al., 2024; Abdelnabi et al., 2024). Moreover, as LLMs participate in data collection, annotation, evaluation, and decision-making, human–LLM ecosystems require interaction protocols that are robust to incentives and strategic reporting (Li et al., 2024; Sun et al., 2024a). Taken together, these developments suggest that the next generation of AI systems will not be shaped by model architectures alone, but by the mechanisms governing how AI agents interact with humans, with data, and with each other (Jordan, 2025). On the other hand, LLMs themselves are increasingly used for mechanism design. Language models can serve as strategic agent proxies, simulators of human behavior, and natural-language interfaces for economic mechanisms, enabling new approaches to mechanism analysis and design in natural language-mediated environments (Shah et al., 2025a; Li et al., 2023; Wu and Hartline, 2024).

In this survey, we develop a unified view of the emerging connection between mechanism design and LLMs. We organize the literature along a bidirectional perspective, as outlined in Figure 1. First, we review LLMs for mechanism design, where language models act as natural-language interfaces for economic mechanisms. Second, we survey mechanism design for LLMs, where incentive-aware principles inform evaluation, alignment, and training in strategic and multi-agent settings. Across both directions, we emphasize that language fundamentally changes how information and feedback are represented. Our goal is to unify these perspectives and use mechanism design as a lens for understanding next-generation AI systems as economic systems that emerge from interacting agents.

## 1.1 Comparison with Existing Surveys

There are several surveys that either focus the intersection of game theory and LLMs (Sun et al., 2025) or focus on specific domains such as LLM-based agents in games (Feng et al., 2025) and LLMs for strategic reasoning and intelligent decision making (Huang et al., 2025b; Zhang et al., 2024). Related overviews include reviews on reward design for LLM alignment (Ji et al., 2025), applications of generative AI in economics (Sajjadi Mohammadabadi et al., 2025) and the intersection of game theory and cybersecurity (Zhu, 2025). In contrast, this survey adopts a mechanism-design perspective and proposes a unified taxonomy of the bidirectional relationship between LLMs and incentive-driven systems. We review both how LLMs are used

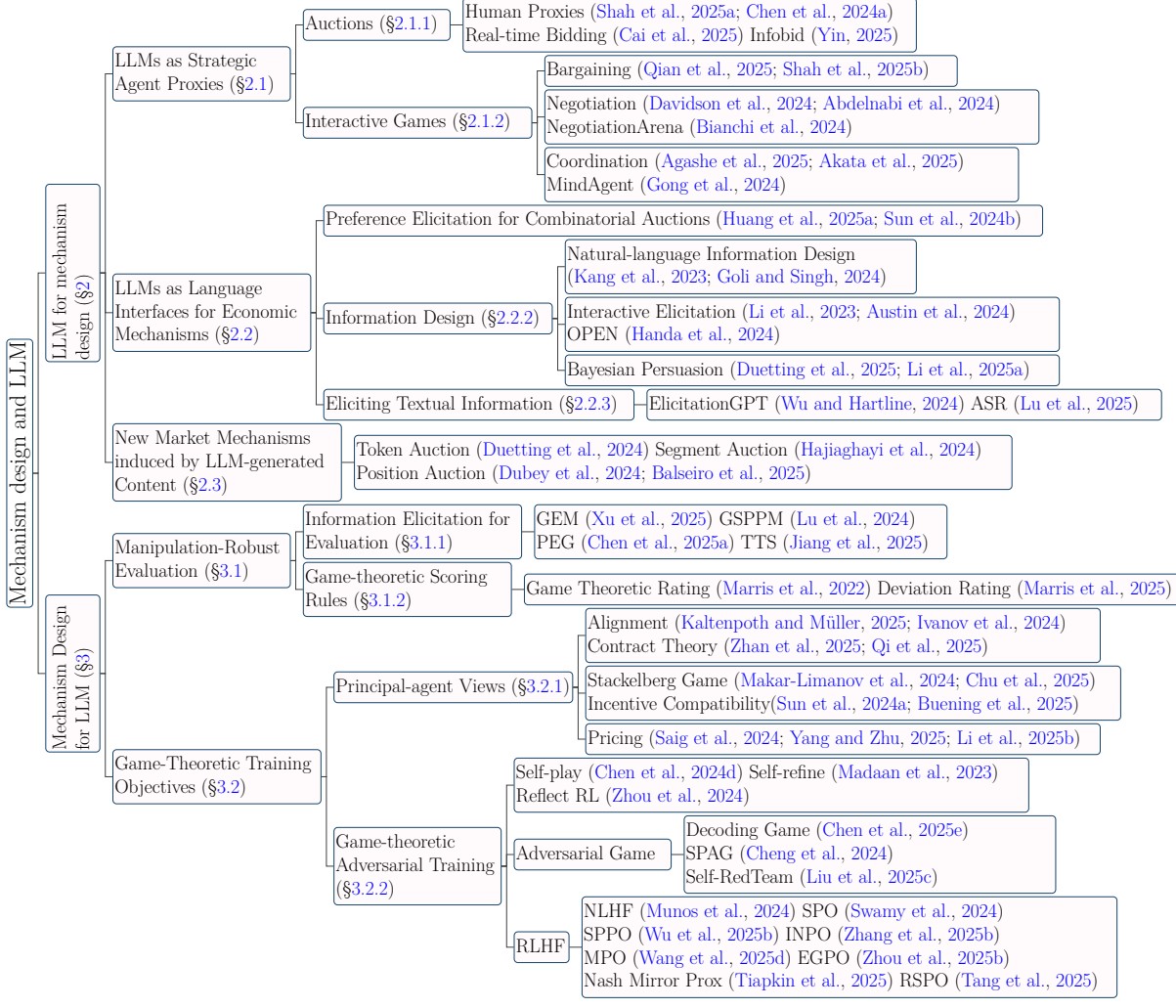

Figure 1: Taxonomy of mechanism design meets LLM.

for mechanism design and how mechanism design principles are applied to LLM evaluation, alignment, and training, highlighting how the two areas inform each other.

The remainder of the paper is organized as follows: Section 2 reviews LLMs for mechanism design. Section 3 discusses mechanism design for LLMs. Section 4 concludes the paper by reaffirming our unified mechanism-design view and its implications for the analysis of LLM-centered systems. Finally, we summarize evaluation protocols, benchmarks, and open-source softwares in Appendix A and outline key open challenges and future directions in Appendix B.

# 2 LLM for Mechanism Design

## 2.1 LLMs as Strategic Agent Proxies

LLMs provide a low-cost, scalable way to simulate strategic agents (Bae et al., 2024; Patel et al., 2024). Their natural language capabilities enable rich communication behaviors (Horton, 2023; Bubeck et al., 2023). We organize this literature into two testbeds: (i) auctions, which study bidding behavior under explicit mechanisms and structured action spaces, and (ii) interactive games, which study bargaining, negotiation, and coordination among LLM agents.

### 2.1.1 Auctions

Auctions are central to efficient resource allocation (Vickrey, 1961), but experimental auction research is often costly, slow, and difficult to replicate (Shah et al., 2025a). These challenges are amplified in dynamic environments such as real-time bidding, where many promising designs remain under-tested or are validated only under simplified settings or limited datasets (Ou et al., 2023). State-of-the-art LLMs (e.g., GPT-4) have demonstrated key auction competencies (Chen et al., 2024a) and often display risk-averse behavior consistent with predictions in strategy-proof environments, enabling a valid proxy for human subjects (Shah et al., 2025a). In highly dynamic settings such as real-time bidding, LLM agents can also reason over context (page content, user segment, budget state), propose bids, and provide explanations for observed trade-offs, supporting both evaluation and design iteration (Cai et al., 2025; Yin, 2025). However, recent work demonstrate challenges and limitations in using LLMs as human proxies (Corrigan et al., 2025). For example, LLM-based economic agents can exhibit systematic behavioral biases that deviate from classical predictions (Fish et al., 2025) and are highly sensitive to model priors and reasoning artifacts (Manning et al., 2024), thereby motivating incentive-aware evaluation and alignment of LLM agents before deploying them in economic environments.

### 2.1.2 Interactive Games: Bargaining, Negotiation and Coordination

LLMs can be modeled as economic agents as they can interpret and produce natural language, and they are trained on large corpora that implicitly encode many economic concepts such as incentives, trade-offs, bargaining norms, and market interactions (Horton, 2023; Bubeck et al., 2023). Compared with classical algorithmic agents that typically require an explicit utility function, structured action space, and carefully specified game rules (Roughgarden, 2010), LLM agents can operate directly in language-mediated environments where preferences and constraints are revealed through dialogue, descriptions, and feedback (Liu et al., 2024). Recent work uses LLMs to play bargaining (Qian et al., 2025; Xia et al., 2024), negotiation (Davidson et al., 2024; Mouri Zadeh Khaki et al., 2025), and coordination games (Agashe et al., 2025; Akata et al., 2025; Gong et al., 2024) in which agents must understand game instructions, maintain conversational context, justify decisions, and adapt strategies over multiple turns; these capabilities make LLMs a natural fit for studying strategic interaction when communication itself is part of the game (Zhang et al., 2024). Empirical evaluations further suggest that LLMs can exhibit elements of economic rationality and microeconomic reasoning (Raman et al., 2024, 2025), responding to incentives, budgets, and opportunity costs, and achieving near-optimal behavior in stylized dynamic market settings. As a result, LLM agents are increasingly used as computational proxies for economic research (e.g., simulating populations in bargaining or coordination tasks), as tools for generating and analyzing economic text, and as assistants for downstream economic decision making in interactive systems (Qian et al., 2025; Wu et al., 2025a; Immorlica et al., 2024). However, current LLMs are highly sensitive to prompt phrasing, yielding behavior that is not sufficiently robust (Sreedhar and Chilton, 2024). A key reason is that most models are not purpose-built for economic reasoning. Targeted post-training and data augmentation can strengthen economic reasoning and stability (Wang et al., 2025c; Zhou et al., 2025c). Moreover, when deploying LLMs in empirical studies, researchers should guard against leakage and

bias by isolating model training data from the study sample and by applying bias corrections alongside uncertainty quantification (Ludwig et al., 2025; Wan et al., 2023).

---

**Discussion on LLMs as Strategic Agent Proxies**

**Key ideas.** LLMs provide a low-cost and scalable proxy for human-subject experiments in auctions and for studying bargaining, negotiation, coordination, and other strategic interactions. This makes LLMs attractive as computational proxies for economic research, policy experiments, and decision-support tools, where they can simulate heterogeneous agents and counterfactual scenarios at scale.

**Challenges.** LLM behavior is highly prompt-sensitive and not purpose-built for economics, raising concerns about robustness, data leakage, and bias. This motivates domain-specialized economic LLMs with targeted post-training and richer economic benchmarks for training and evaluation.

---

## 2.2 LLMs as Language Interfaces for Economic Mechanisms

This section treats LLMs as language interfaces for economic mechanisms and groups the literature into four roles: (i) preference elicitation, where dialogue is mapped into structured reports (e.g., bids) for mechanisms such as combinatorial auctions; (ii) language-based information design/persuasion, where textual outputs shape beliefs and strategic behavior; and (iii) text elicitation with incentives, where private information is inherently free-form text and must be scored or aggregated reliably.

### 2.2.1 Preference Elicitation for Auctions

Combinatorial auctions face a fundamental expressiveness tension: fully expressive bidding allows bids on any subset of items and therefore requires reporting values for exponentially many bundles, namely $2^m$ bundles for $m$ items, leading to computational intractability (Nisan, 2000). Recent work explores LLMs as natural-language interfaces for preference elicitation, aiming to reduce reporting frictions without sacrificing allocation quality (Soumalias et al., 2025). Rather than asking bidders to enumerate bundle values, LLMs can conduct interactive dialogues to extract constraints, priorities, and trade-offs, and then compile the resulting information into structured bids in a compact language that a mechanism can optimize over (Huang et al., 2025a). In parallel, LLMs can incorporate semantic context into bidder-specific valuation models to denoise imprecise descriptions and improve value estimates from partial reports (Sun et al., 2024b). Together, these approaches lower communication costs while targeting high-welfare outcomes in practical combinatorial allocation settings (Soumalias et al., 2025; Shah et al., 2025a).

### 2.2.2 Natural-Language Information Design and Bayesian Persuasion

Information design studies how a principal can strategically structure and disclose information to influence the decisions of Bayesian agents in a way that improves the designer's objective (Bergemann and Morris, 2019). A central prerequisite in such problems is how agents value different outcomes; without preference information, it is impossible to construct information

structures that meaningfully shape behavior (Bergemann and Morris, 2005; Falk et al., 2018). For example, bidder preferences are pivotal in combinatorial auctions (Huang et al., 2025a); recommender systems infer user preferences to personalize content and support downstream decision-making (Austin et al., 2024; Kang et al., 2023); in modern LLM pipelines, preference data is foundational to reinforcement learning from human feedback (Ouyang et al., 2022; Cai and Li, 2025). Traditional preference elicitation methods rely on report-based mechanisms where agents disclose valuations or types and the system infers optimal outcomes Chen and Pu (2004); Kang et al. (2023). Although such elicitation mechanisms are theoretically query-efficient, the required communication can still be cognitively demanding (Huang et al., 2025a). LLMs offer new advantages through natural language understanding and generation, allowing them to capture implicit and context dependent preferences that elude traditional structured approaches (Kang et al., 2023; Goli and Singh, 2024; Ronanki et al., 2023).

To obtain more accurate preference information, LLMs can engage users in interactive multi turn dialogue to explore open questions and elicit task specific preferences (Li et al., 2023). Compared with traditional user written prompts or static labels, this conversational approach better captures the complexity and implicit intentions of human preferences (Kendapadi et al., 2025). For recommender systems, For recommender systems, LLMs can generate a semantic summary after each interaction to reflect the system's current understanding of the user's preferences, allowing users to verify or correct this summary (Loepp and Ziegler, 2024) and can proactively ask clarifying questions to iteratively refine the preference estimate Montazeralghaem et al. (2025). These approaches leverage the ability of LLMs to engage in interactive conversations, using in-context learning to accurately infer user preferences. Building on this foundation, another line of work focuses on optimizing the multi turn interaction to explore the preference space and fully learn user preferences. Using Bayesian optimization, the system selects the most informative natural language questions with uncertainty driven decision strategies, enabling more efficient exploration and better preference learning Austin et al. (2024); Handa et al. (2024).

LLMs not only help elicit and interpret preferences, but also act as language-based information designers that structure and design information to shape beliefs and guide decisions (Lin et al., 2025).In particular, in Bayesian persuasion settings, LLMs can design information that is easier to understand and more effective at influencing beliefs and downstream choices (Duetting et al., 2025). Their linguistic fluency enables a systematic exploration of rich framing strategies that were previously inaccessible in traditional models (Cheng and You, 2025). Building on this perspective, Li et al. (2025a) design practical systems that implement Bayesian persuasion mechanisms directly in natural language, while Wu et al. (2025a) develop methods for generating persuasive marketing contents.

### 2.2.3 Eliciting Textual Information

LLMs can generate rich text such as comments, reviews, and summaries (Chen et al., 2024b). In many applications, however, the central challenge how to elicit informative and truthful textual reports: there is often no objective ground truth, and the underlying criteria are subjective, so naive incentives reward fluency, verbosity, or strategic hedging rather than genuine information (Schneider et al., 2024; Lu et al., 2025). Classical proper scoring rules provide a principled

foundation for truthful information elicitation by making honest belief reporting a dominant strategy, but they rely on numerical forecasts and realized outcomes (Gneiting and Raftery, 2007). Directly applying these ideas to open-ended text is infeasible, motivating mechanisms that operate over semantic representations rather than surface forms. ElicitationGPT (Wu and Hartline, 2024) addresses this challenge by mapping textual reports into a learned semantic space and applying a semantic proper scoring rule under a know-it-or-not model. This design incentivizes agents to accurately express what they know and to explicitly acknowledge uncertainty, rather than optimizing for stylistic plausibility. Aligned Scoring Rules (ASR) (Lu et al., 2025) extend this approach by adapting the semantic scoring mechanism to task-specific alignment objectives and preference distributions, preserving truthful elicitation guarantees while improving robustness and flexibility across domains.

> **Discussion on LLMs as Language Interfaces**
>
> **Key ideas.** Since LLMs can parse free-form language and run interactive, g dialogues effectively, it is possible to elicit richer and more structured preferences and feed them into preference-learning pipelines. This enables more expressive mechanisms for resource-allocation problems.
>
> **Challenges.** Current models struggle with preferences expressed over long, multi-turn interactions, making their inferred preferences noisy and challenging (Zhao et al., 2025). A key need is high-quality datasets of real human preferences that capture this complexity, to support reliable preference learning and mechanism design.

## 2.3 New Market Mechanisms Induced by LLM-Generated Contents

LLM-generated contents create two concrete scarce resources: generation budget during decoding and visibility in ranked outputs (Feizi et al., 2024). These constraints motivate auction designs that price either generation resources or exposure directly. Accordingly, we focus on two new-arisen mechanisms: token auctions (Duetting et al., 2024), which allocate and price tokens or decoding steps, and position auctions (Balseiro et al., 2025), which allocate visibility in LLM-mediated rankings such as answers, summaries, or recommendations.

### 2.3.1 Token Auctions

The rise of LLMs opens new avenues for auction design because the model can now participate directly in how ads and organic content are composed. At one extreme, a token auction treats each advertiser as an LLM that proposes a next-token distribution; the mechanism aggregates these multiple LLM outputs into a single continuation using bid-dependent weights (Duetting et al., 2024). A related approach aggregates advertisers' LLM-encoded preferences while preserving incentive compatibility (Soumalias et al., 2024). A weaker form of integration keeps the LLM's organic answer fixed and fuses it with an advertisement. Advertisers then bid for this sponsored answer, and the platform selects the winner (Mordo et al., 2024). Moving from whole-answer fusion to finer granularity, segment auctions use retrieval-augmented generation (RAG) to score how relevant each ad is to individual sentences or paragraphs, and then choose one or

more ads per segment via a truthful randomized rule that maximizes a logarithmic social-welfare objective (Hajiaghayi et al., 2024).

### 2.3.2 Position Auctions

Once ads are interwoven with generated content, even the logic of classic position auctions becomes unstable. Traditional positions have fixed prominence, but LLM-generated layouts make exposure stochastic and context-dependent (Varian, 2007; Dubey et al., 2024). The auction can no longer determine display outcomes on its own; it must coordinate with the LLM that renders the final arrangement (Dubey et al., 2024). Moreover, a position's value is no longer common across advertisers: the utility of an ad depends on surrounding text, and attention-based user models can predict clicks when multiple creatives are shown jointly, enabling mechanisms that optimize over full joint layouts (Balseiro et al., 2025).

---

**Discussion on New Market Mechanisms**

**Key ideas.** This section discusses two ways in which LLMs reshape mechanism design. First, token-level auctions redefine the unit of trade, shifting attention from whole items to finer-grained units such as tokens or generation segments. Second, position auctions become increasingly relevant in LLM-mediated settings, where ranked exposure and attention over generated content play a central role.

**Challenges.** Designing mechanisms for LLM generated content raises challenges beyond classical settings. LLM outputs are context dependent, stochastic, and uncertain in quality, so mechanisms should explicitly incorporate and adapt to this uncertainty. In addition, matching based designs that assign queries to models, prompts, or human oversight remain underexplored and offer a promising direction.

---

## 3 Mechanism Design for LLMs

### 3.1 Manipulation-Robust Evaluation

LLMs have exceptional text-generation ability, such as generating comments, reviews, and summaries (Chen et al., 2024b). Yet, evaluating these LLM outputs remains difficult primarily because ground truth is often unavailable: the tasks are inherently subjective, and obtaining consistent human judgments is costly (Schneider et al., 2024). LLMs-as-judges offer a scalable solution. However, their judgments can deviate from human references, and their evaluations usually lack provable guarantees, leaving them vulnerable to strategic manipulation (Lu et al., 2025).

### 3.1.1 Information Elicitation for Evaluation

Peer prediction mechanisms offer an alternative elicitation paradigm that guarantee more information is scored higher in expectation (Miller et al., 2005). A key challenge of applying this idea to textual data is how to predict a textual response $X_i$ given another textual response $X_j$ (e.g. $X_i$ and $X_j$ can be two reviews of the same paper). Lu et al. (2024) first propose a solution by

using an open-source LLM as a textual predictor, where the probabilities $\Pr(X_j \mid X_i)$ can be estimated by applying the chain rule on the next-token-prediction probabilities of the open-source LLM. This probability can then be used to compute an information score for incentivizing high-effort signal $X_i$. Xu et al. (2025) then adapt the above idea and design an evaluator that directly estimates the pointwise mutual information between two agents' responses, which additionally estimates $\Pr(X_j)$ and removes the prior information. The pointwise mutual information score then serves as a verification-free and manipulation-resistant benchmark for evaluating LLMs' judgment abilities. The above two methods use an open-box LLM as a textual predictor, while there are other methods that use a strong LLM as an oracle (closed-form adaptation). For example, Chen et al. (2025a) adapt multi-task peer prediction for answer evaluation in order to incentivize more informative and truthful responses. Such mechanisms can be further integrated into retrieval and generation pipelines to elicit higher-quality evidence and produce more informative textual reports (Jiang et al., 2025). Furthermore, Zhang et al. (2025a) show that peer prediction can also be used to score residual human information in addition to LLM, which can be used to incentivize human effort and detect LLM cheaters. However, their method primarily applies to labeling settings.

### 3.1.2 Game-theoretic Scoring Rules

Traditional AI evaluation methods typically assume that candidate models can be meaningfully ordered along a single dimension of strength. Representative approaches such as Elo (Elo and Sloan, 1978) and the Bradley–Terry model (Bradley and Terry, 1952) aggregate pairwise outcomes into a single global ranking, forming the foundation of popular crowdsourced LLM benchmarks such as Chatbot Arena (Chiang et al., 2024). However, the single-dimensional assumption is increasingly fragile in modern settings, where AI capabilities are highly multidimensional, interactions are strategic, and benchmarks are dynamic rather than fixed (Marris et al., 2022). To address these challenges, Liu et al. (2025d) reframes open-ended LLM evaluation as a multi-player game and replaces Elo with clone-invariant, equilibrium-based ratings, improving robustness to prompts redundancy. Complementing this perspective, deviation ratings (Marris et al., 2025) formalize redundancy robustness as a property of the scoring rule itself: duplicating or near-duplicating strategies or prompts should not change the resulting ratings, preventing redundancy from distorting evaluation.

---

**Discussion on Mechanism for Evaluation**

**Key ideas.** This section discusses how to adapt peer prediction and scoring rules for data evaluation. Together, these methods offer alternatives to align LLM-based evaluators with human judgments in settings such as grading, review aggregation, and content moderation. **Challenges.** A key direction is to integrate these techniques into end-to-end LLM pipelines, covering human annotation, RAG-based retrieval and summarization, and downstream deployment so that truthfulness incentives and robustness guarantees hold throughout the full workflow.

---

## 3.2 Game-theoretic Training Objectives

The cornerstone of the success of LLMs is alignment, which enables them to generate text that reflects human preferences for both usability and safety (Köbis et al., 2025). At the same time, user preferences are heterogeneous across tasks and individuals (Yang et al., 2019; Conitzer et al., 2024). This heterogeneity motivates incentive-aware alignment frameworks, which we group into (i) principal–agent, contract-theoretic approaches that design explicit incentives for truthful, high-effort feedback, and (ii) game-theoretic adversarial training approaches that model strategic interactions to improve robustness and empirical performance.

### 3.2.1 Principal-Agent Views

Contract theory studies how a principal can optimally design rewarding schemes to align an agent's actions with the principal's objectives under information asymmetry (Bolton and Dewatripont, 2004), thus offering a principled toolkit for preference alignment and the incentivization of LLM agents toward desirable outcomes (Kaltenpoth and Müller, 2025). This can be modeled as a multi-turn reinforcement learning, where the principal updates a sequence of contracts to steer behavior, and the agent learns a policy that targets social welfare (Ivanov et al., 2024). Complementing this, Zhan et al. (2025) use online contract design to infer hidden provider choices and to compute bonus schemes that reliably induce high-quality AI-generated content. This framework also applies to multi-agent systems, where smart contracts reward truthful behavior and ensure transparent collaboration (Qi et al., 2025).

Overall, contract-theoretic work treats alignment as explicit incentive design to steer behavior and elicit reliable feedback. In contrast, Stackelberg formulations emphasize strategic response dynamics between the policy and a learned (or adversarial) reward model, casting alignment as a bilevel game (Makar-Limanov et al., 2024). In some variants of the framework, the follower is modeled as an adversarial preference distribution to increase robustness to annotation noise and distribution shift (Chu et al., 2025). In multi-objective alignment, diverse groups or labelers may strategically misreport to steer the outcome and reduce overall performance. A VCG-style payment mechanism (Sun et al., 2024a) or an approximately strategyproof algorithm (Buening et al., 2025) can both encourage truthful reporting and prevent misreporting that would bias the aggregated objective (Manyika et al., 2025). In LLM systems, these mechanisms are implemented by aggregating user or labeler feedback into the fine-tuning objective (e.g., RLHF/DPO) and assigning rewards or credits based on each agent's marginal contribution or strategyproof aggregation rules, making misreporting unprofitable.

Another challenge is that LLM services are often black boxes priced per token or per call (Velasco et al., 2025). This pricing rewards verbosity and allows providers to quietly lower effort while charging the same, creating a moral hazard. A contract-theoretic remedy is to tie payment to verifiable outcomes, such as accuracy, which has been shown to improve generation quality (Saig et al., 2024). The same idea extends to multi-dimensional menus that price speed, accuracy, reliability, and liability, so users choose what they value and providers are incentivized to deliver it (Yang and Zhu, 2025). In parallel, Li et al. (2025b) design usage-based prices that incorporate users' heterogeneous prompt capabilities and strategic behaviors, thereby maximizing platform payoff while promoting efficient service utilization.

### 3.2.2 Game-Theoretic Adversarial Training

Game-theoretic modeling formalizes LLMs as agents in strategic interactions and motivates empirical game–based adversarial training for improved learning and empirical performance. LLM self play seeks to build strong agents with minimal human oversight via reflection, critique, and iterative improvement (Fu et al., 2023). One approach is self-reflection at the prompt level that repeatedly refines reasoning (Chen et al., 2024d; Madaan et al., 2023; Zhou et al., 2024). Another approach frames self-play as an adversarial game in which one agent generates challenging inputs while the other attempts to solve them (Chen et al., 2025b), often as attacker and defender pairings (Cheng et al., 2024; Liu et al., 2025c). These formulations can also help improve prompts through strategic interaction by generating optimal prompts under adversarial pressure (Ye et al., 2025; Chen et al., 2025e).

Motivated by this adversarial perspective, recent work frames alignment as a two-player min–max game between policies and proposes algorithms grounded in Nash equilibria (Ouyang et al., 2022; Ge et al., 2024; Rosset et al., 2024). For example, NLHF models alignment as a two-player zero-sum preference game where each policy aims to outperform the other under a learned preference function (Munos et al., 2024). Follow-up studies focus on training stability and robustness (Swamy et al., 2024; Wu et al., 2025b; Zhang et al., 2025b), as well as on providing theoretical guarantees and improved interpretability, for example by designing more robust gradient methods with better convergence properties(Wang et al., 2025d; Tiapkin et al., 2025; Zhou et al., 2025b). Additionally, KL-regularization during policy optimization further stabilizes learning and yields generalization benefits (Ye et al., 2024; Tang et al., 2025).

> **Discussion on Game-Theoretic Training**
>
> **Key ideas.** The principal–agent framework models LLM alignment as a game between a principal and an agent, enabling algorithms that learn equilibrium policies. Contract theory then provides incentive-compatible designs, strengthening both the theoretical foundations and interpretability of alignment procedures.
>
> **Challenges.** Most existing work focuses on a single-principal, single-agent setting, while real deployments involve multiple principals and agents with heterogeneous objectives, competition, and externalities. Developing stronger guarantees and scalable algorithms for these multi-agent, multi-principal environments on equilibrium existence/uniqueness, convergence rates, and partial observability remains an important open problem.

## 4 Conclusions

In this survey, we develop a unified mechanism-design view of the bidirectional relationship between LLMs and incentive-driven systems. LLMs increasingly operate within economic mechanisms, while mechanism design provides principled tools to evaluate, align, and train LLMs in strategic multi-agent settings with heterogeneous, manipulable feedback. Together, we view next-generation AI systems as economic systems shaped by interactions among models, users, platforms, and data providers, making mechanism design a unifying lens for their analysis and design. Open challenges and future directions are deferred to Appendix B.

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

# A    Benchmarks and Softwares

Using LLMs as tools for mechanism design requires understanding how well they engage in strategic reasoning. Current empirical evaluations come from a diverse set of benchmarks that emphasize different sources of strategic complexity, ranging from opponent modeling to social interaction to incentive-driven behavior. To clarify what is known and identify what remains untested, we review existing evaluations of LLM strategic behavior and organize them along three dimensions of strategic complexity: (1) individual strategic reasoning in simple games, (2) social strategic interaction in multi-agent settings involving cooperation, coordination, or deception, and (3) incentive-driven strategic behavior in economic environments such as bargaining, persuasion, and market reasoning. Table 1 provides a summary of representative benchmark results across three areas, and Table 2 summarizes representative software implementations and simulation platforms.

Given the rise of agentic AI, a growing line of benchmarks evaluates LLMs' strategic reasoning in simple games (Akata et al., 2025). Several works provide relatively comprehensive coverage over diverse games. For example, GTBench (Duan et al., 2024) tests LLMs on board games, card games, and a range of other classical games; Wu et al. (2024a) study performance on six games, including rock–paper–scissors and the MinEscort game; and Wang et al. (2025b) introduce a systematic benchmark covering 144 two-player games. Other efforts focus on specific game types. Hu et al. (2025) evaluate LLMs in video-game environments, Topsakal et al. (2024) consider grid-based games, and ClemBench (Chalamalasetti et al., 2023) focuses on dialogue games that test whether LLMs can reliably follow game instructions (Abdelnabi et al., 2024). A complementary line of work targets social deduction games, including Hoodwinked (O'Gara, 2023), Werewolf (Xu et al., 2024), and Avalon (Light et al., 2023).

The second line of benchmarks extends simple-game scenarios to multi-agent settings, where collaboration, coordination, competition, and deception can emerge (Liang et al., 2025; Tamer and Gümüş, 2025; Mordo et al., 2025). An important strand focuses on negotiation behavior, such as GameBench (Hua et al., 2024) and NegotiationArena (Bianchi et al., 2024). Other benchmarks emphasize cooperation, including Welfare Diplomacy (Mao et al., 2024; Mukobi et al., 2023; , FAIR), which builds on Diplomacy and requires alliance formation and joint planning, and Minecraft-based environments (Wang et al., 2025a; Gong et al., 2024; Fan et al., 2022; Schipper et al., 2025), which test LLMs' ability to coordinate with other agents on long-horizon, open-ended tasks. Complementing these environment-specific benchmarks, LLMArena (Chen et al., 2024c) and LLMsPark (Chen et al., 2025c) provide multi-agent, game-theoretic evaluation frameworks across several dynamic environments, targeting abilities such as opponent modeling, communication, team collaboration, and strategic depth (Yin et al., 2025).

Lastly, some empirical evaluations focus on economic behavior, revealing elements of economic rationality (Raman et al., 2024) , microeconomic reasoning (Raman et al., 2025) and bounded rationality (Liu et al., 2025a). There are also evaluations based on a variety of economic scenarios, including auction games (Chen et al., 2024a), which study how LLM agents bid and respond to auction rules; bargaining games (Qian et al., 2025), which model multi-round negotiation over surplus division; persuasion games (Idziejczak et al., 2025; Karande et al., 2024; Shapira et al., 2025), where an expert agent attempts to influence a decision-maker through

| Strategic Dimension | Benchmark | Environment / Game Type |
|---|---|---|
| Individual Strategic Reasoning | GTBench (Duan et al., 2024) | Game-theoretic games |
| | SmartPlay (Wu et al., 2024b) | Strategic, planning, and embodied games |
| | TMGBench (Wang et al., 2025b) | 144 two-player games |
| | ClemBench (Chalamalasetti et al., 2023) | Dialogue-based instruction-following games |
| | LMGAME-Bench (Hu et al., 2025) | Video-game environments |
| | LLM-Deliberation (Abdelnabi et al., 2024) | Scorable multi-issue negotiation games |
| Social Strategic Interaction | GameBench (Hu et al., 2025) | Multi-agent negotiation games |
| | CuisineWorld (Gong et al., 2024) | Multi-agent collaboration games |
| | PillagerBench (Schipper et al., 2025) | Competitive multi-agent games |
| | LLMARENA (Chen et al., 2024c) | Multi-agent dynamic games |
| | LLMsPark (Chen et al., 2025c) | Game-theoretic multi-agent games |
| | WGSR-Bench (Yin et al., 2025) | Wargame-based strategic games |
| Incentive-Driven Economic Behavior | STEER (Raman et al., 2024) | Economic decision tasks |
| | STEER-ME (Raman et al., 2025) | Microeconomic reasoning tasks |
| | CHBench (Liu et al., 2025a) | Auction games |
| | Among Them (Idziejczak et al., 2025) | Social deception games |

Table 1: Representative benchmarks for evaluating strategic reasoning in LLMs, organized by strategic dimension.

messages; and marketplace environments such as Magentic-Marketplace (Bansal et al., 2025), which study two-sided agentic markets in which assistant agents represent consumers and service agents represent competing businesses, allowing the analysis of welfare, bias, and search frictions. Finally, Guo et al. (2024a) propose an economic arena based on competitive games and empirically analyze how LLM agents' play converges (or fails to converge) to Nash equilibria. Additionally, researchers also train domain-specific LLMs for economics and finance, using curated corpora and post-training to enhance economic reasoning and better align agent behavior with formal models (Zhou et al., 2025c).

# B   Open Challenges and Future Directions

**Capability Mismatch in LLM-based Mechanism Design** Despite their strong generative and reasoning capabilities, LLMs are not purpose-built for mechanism design tasks such as auctions, incentive alignment, or equilibrium reasoning, and may therefore miss key economic constraints without explicit structure or supervision (Zhou et al., 2025c). Moreover, mechanism design is inherently multi-agent: outcomes depend on induced incentives and strategic interaction networks, motivating structure-aware and game-theoretic modeling of LLM-based systems (Zhou et al., 2025a).

**Human–AI Collaboration and Incentive Alignment** As LLM agents move from tools to decision-making partners, human-AI systems must account for heterogeneous objectives, costs, and strategic behavior (Frankel, 2014). This motivates incentive-aware interaction protocols that

| Software / Framework | System Type | Primary Capability |
|---|---|---|
| Hoodwinked (O'Gara, 2023) | Social deduction environment | Deception and cooperation analysis in text-based social games |
| Werewolf (Xu et al., 2024) | Social deduction environment | Strategic reasoning and deception in multi-agent settings |
| Avalon (Light et al., 2023) | Social deduction environment | Trust and deception analysis in team-based games |
| Alympics (Mao et al., 2025) | Game-theoretic simulation framework | Strategic interaction and robustness analysis |
| NegotiationArena (Bianchi et al., 2024) | Multi-agent negotiation framework | Communication and negotiation strategy analysis |
| VOYAGER (Wang et al., 2025a) | Embodied agent framework | Lifelong skill learning and compositional generalization |
| MindAgent (Gong et al., 2024) | Multi-agent coordination framework | Planning and collaboration infrastructure |
| MINEDOJO (Fan et al., 2022) | Embodied agent framework | Open-ended task learning in Minecraft environments |
| TactiCrafter (Schipper et al., 2025) | Multi-agent system | Tactical coordination and adaptive learning |
| AucArena (Chen et al., 2024a) | Auction simulation environment | Strategic bidding and adaptive decision-making analysis |
| Persuasive Games with LLMs (Karande et al., 2024) | Multi-agent persuasion framework | Personalized persuasion and resistance-aware dialogue |
| Magentic Marketplace (Bansal et al., 2025) | Agentic marketplace simulation | Market dynamics and strategic behavior analysis |
| Economics Arena (Guo et al., 2024a) | Competitive game simulation | Strategic reasoning and rationality analysis |

Table 2: Representative software frameworks and simulation environments used to study strategic, social, and economic behavior of LLM agents.

elicit truthful information, allocate tasks efficiently, and mitigate manipulation or misalignment in real deployments (Wu and Hartline, 2024; Buening et al., 2025).

**LLM for Explaining Mechanisms** LLMs open new opportunities for explaining and interacting with mechanisms (Orner et al., 2025). Recent work uses LLMs to produce natural-language rationales for outcomes and to frame automated mechanism design as code generation of interpretable implementations (Liu et al., 2025b; Hosseini and Khanna, 2025). A key direction is to build faithful explanation frameworks for voting, matching, and auctions by grounding natural-language interfaces.

**Market Design for LLMs.** As LLMs become stand-alone products, platforms must design markets to allocate limited inference capacity and price access (Feizi et al., 2024). Emerging mechanisms in online advertising include position auctions and token auctions. Related questions arise in data markets (Castro Fernandez, 2023), recommender systems (Yao et al., 2024), and optimal pricing of AI services (Bergemann et al., 2025).

# C   Experiments

We evaluate the ability of using LLMs as proxies for human participants in sealed bid auction experiments. Following Shah et al. (2025a), we simulate auctions with three homogeneous LLM agents bidding against one another. Each simulation proceeds through six stages: (1) planning, where agents interpret the auction rules and objectives; (2) value realization, in which private

valuations are drawn or assigned; (3) bidding, where agents submit bids based on their strategies; (4) outcome determination, including allocation and payments; (5) reflection, where agents receive feedback and update their reasoning; and (6) repetition, allowing behavior to evolve over multiple auction rounds. We evaluate three cutting-edges LLMs: GPT5.1, GPT4omini and Gemini2.5 (et al., 2024; Comanici et al., 2025).

## C.1 Setting

We study four sealed-bid auction formats: First-Price (FPSB), Second-Price (SPSB), Third-Price (TPSB), and All-Pay in an independent private values (IPV) environment. Each auction features three bidders. Bidder $i$'s private valuation is drawn independently from a uniform distribution,

$$v_i \sim \text{Unif}\{0, \ldots, 99\}.$$

After observing $v_i$, bidder $i$ submits a sealed bid $b_i = \beta_i(v_i)$, where $\beta = (\beta_1, \beta_2, \beta_3)$ denotes the bidding strategy profile mapping values to bids.

In the FPSB auction, the highest bidder wins the object and pays her own bid, while all other bidders pay zero. The payment rule is

$$t_i(b) = \mathbf{1}\{i \text{ wins}\} \cdot b_i.$$

In the SPSB auction, the highest bidder wins and pays the second-highest bid:

$$t_i(b) = \mathbf{1}\{i \text{ wins}\} \cdot b_{(2)},$$

where $b_{(2)}$ denotes the second-order statistic (the second-highest bid).

In the TPSB auction, the highest bidder wins and pays the third-highest bid:

$$t_i(b) = \mathbf{1}\{i \text{ wins}\} \cdot b_{(3)},$$

where $b_{(3)}$ denotes the third-order statistic (the third-highest bid).

Finally, in the all-pay auction, all bidders pay their bids regardless of the outcome, and the highest bidder wins the object:

$$t_i(b) = b_i.$$

Bids are restricted to integer-dollar amounts (i.e., $1 increments), and ties are broken uniformly at random. For each of the auction formats, we perform 3 experiments with 10 rounds and 3 LLM bidders competing over a prize.

## C.2 Baselines

The risk-neutral equilibrium behavior in these four auction formats is well understood. Table 3 summarizes the equilibrium bidding functions specialized to our setting with $n = 3$ bidders and i.i.d. values $v_i \sim \text{Unif}\{0, \ldots, 99\}$. A single asterisk ($*$) indicates a Bayes–Nash equilibrium, whereas a double asterisk ($**$) denotes a dominant-strategy equilibrium. In terms of incentive compatibility, only the second-price auction is (dominant-strategy) incentive compatible:

| Auction Type | Risk-Neutral Equilibrium Strategy |
|---|---|
| First-Price Sealed-Bid* | $b(v_i) = \frac{2}{3}v_i$ |
| Second-Price Sealed-Bid** | $b(v_i) = v_i$ |
| Third-Price Sealed-Bid* | $b(v_i) = 2v_i$ |
| All-Pay Auction* | $b(v_i) = \frac{2}{3 \cdot 99^2} v_i^3$ |

Table 3: Risk-neutral equilibria in IPV auctions with three bidders and values $v_i \sim \mathrm{Unif}\{0, \ldots, 99\}$. A single asterisk ($*$) denotes a Bayes–Nash equilibrium, while a double asterisk ($**$) denotes a dominant-strategy equilibrium.

| | Distance to Bayes-Nash Equilibrium | | |
|---|---|---|---|
| Auction | GPT5.1 | GPT4o-mini | Gemini2.5flash |
| First-price | $8.47_{\pm 7.64}$ | $17.84_{\pm 15.17}$ | $13.57_{\pm 10.32}$ |
| Second-price | $11.37_{\pm 6.35}$ | $11.98_{\pm 14.67}$ | $6.98_{\pm 8.18}$ |
| Third-price | $61.52_{\pm 27.90}$ | $53.89_{\pm 30.28}$ | $48.33_{\pm 25.58}$ |
| All-pay | $15.80_{\pm 13.02}$ | $31.14_{\pm 15.34}$ | $24.15_{\pm 15.26}$ |
| | Distance to Incentive-Compatible Behavior | | |
| Auction | GPT5.1 | GPT4o-mini | Gemini2.5flash |
| First-price | $17.00_{\pm 8.91}$ | $19.90_{\pm 17.13}$ | $8.42_{\pm 8.75}$ |
| Second-price | $11.37_{\pm 6.35}$ | $11.98_{\pm 14.67}$ | $6.98_{\pm 8.18}$ |
| Third-price | $11.82_{\pm 7.92}$ | $15.72_{\pm 19.82}$ | $10.06_{\pm 14.68}$ |
| All-pay | $19.33_{\pm 10.41}$ | $8.14_{\pm 9.09}$ | $11.91_{\pm 10.75}$ |

Table 4: Distance between LLM bidding behavior and theoretical benchmarks.

truthful bidding $b_i(v_i) = v_i$ is optimal regardless of others' bids.

## C.3 Results

We compare each LLM's bidding behavior against the risk-neutral equilibrium and the incentive-compatible benchmark in Figures 2, 3, and 4. For FPSB, most bids exceed the Bayes-Nash equilibrium, consistent with empirical findings in (Shah et al., 2025a) and (Cox et al., 1988). Relative to GPT-5.1, GPT-4o mini shows more structured departures from the equilibrium curve, including non-monotonicity and locally inflated bids at lower valuation levels, plausibly reflecting its more limited capacity. For SPSB, both GPT-5.1 and GPT-4o mini tend to underbid, whereas Gemini 2.5 is closer to the equilibrium benchmark. All three models underbid substantially in TPSB, while the all-pay setting exhibits pronounced overbidding, reflecting its greater strategic complexity. We summarize the deviations between LLM bidding behavior and theoretical benchmarks in Table 4. Together, these results underscore the need to design LLM agents with domain-specific structure and supervision tailored to mechanism design.

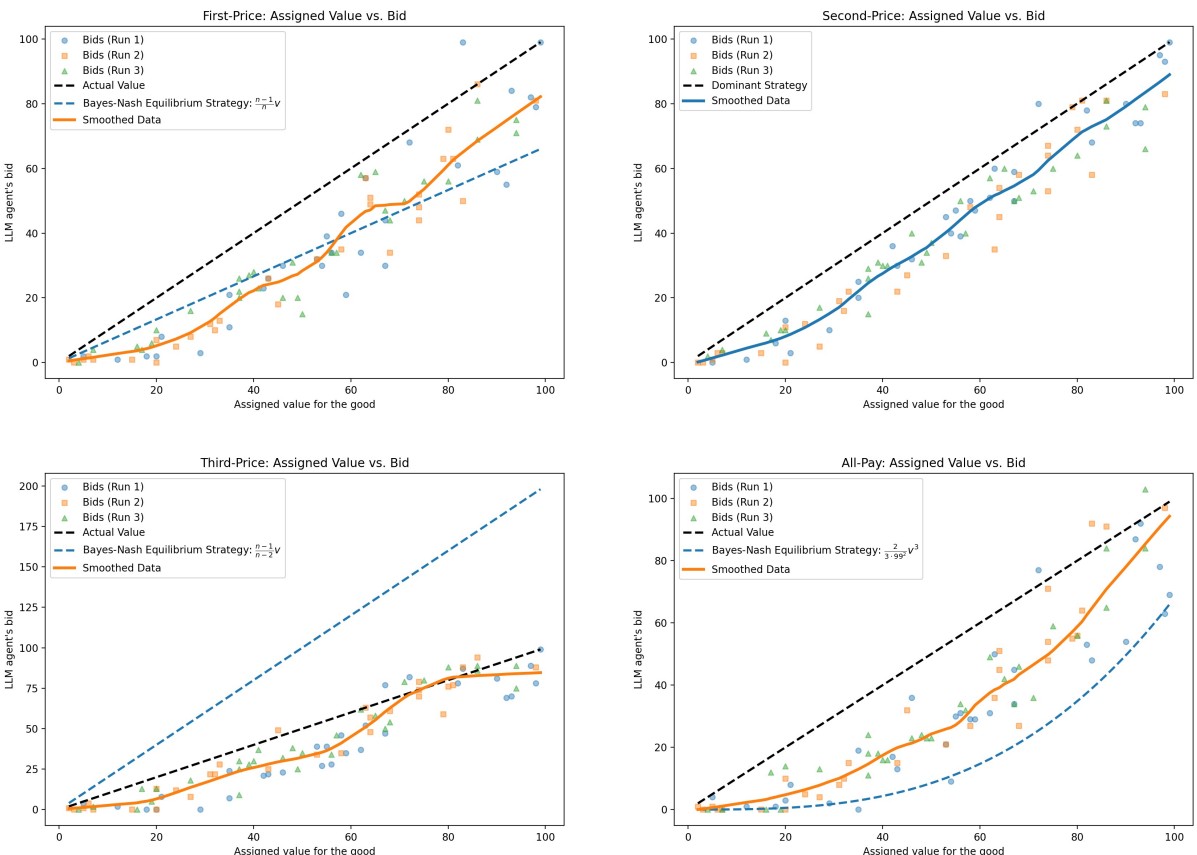

Figure 2: **GPT-5.1 bidding behavior across four auction formats under the IPV setting.** Bidders' private values are independently drawn from the uniform distribution on $\{0, \dots, 99\}$, and GPT-5.1 bids are plotted against values (blue circles, orange squares, and green triangles). The black dotted line indicates truthful bidding $b(v) = v$ (the dominant strategy in second-price auctions). The blue dashed line shows the Bayes–Nash equilibrium bidding strategy reported in Table 3. The solid orange line is the LOESS-smoothed bidding curve.

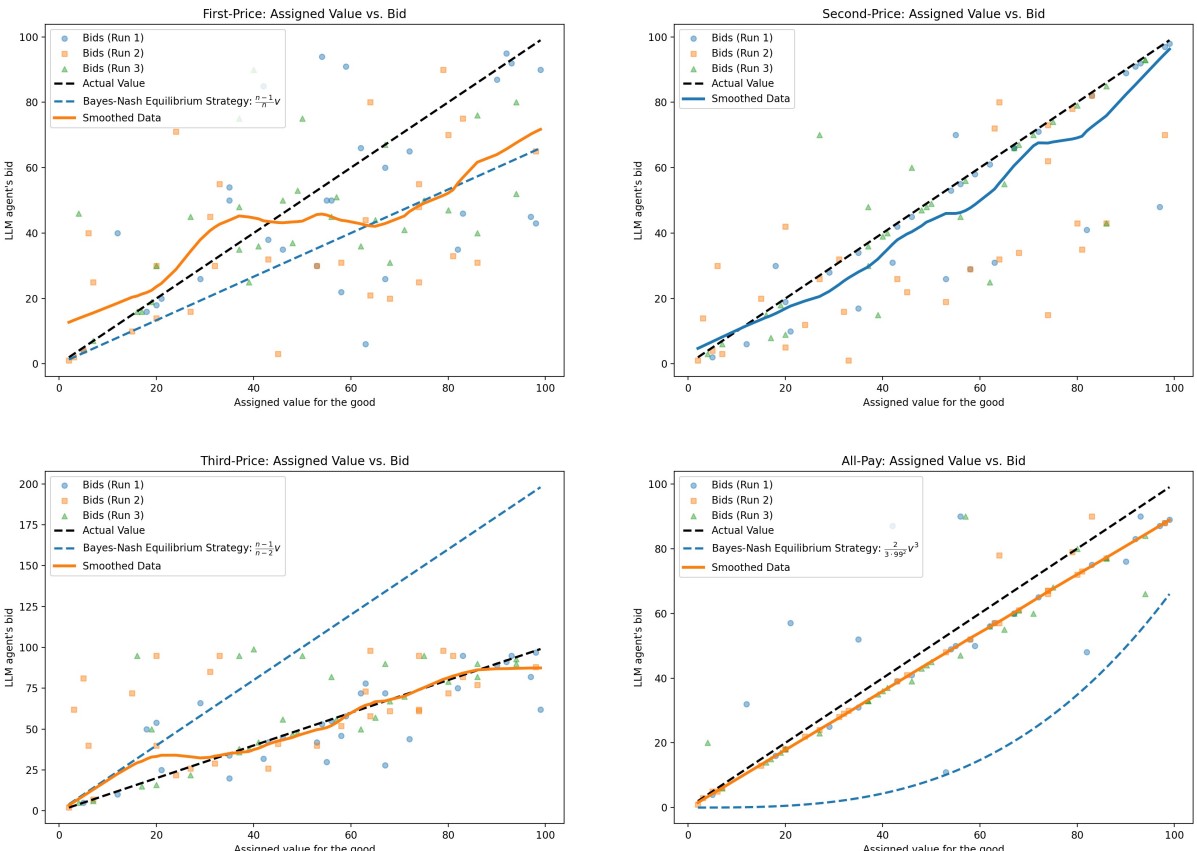

Figure 3: **GPT-4omini bidding behavior across four auction formats under the IPV setting.** Bidders' private values are independently drawn from the uniform distribution on $\{0, \ldots, 99\}$, and GPT-4omini bids are plotted against values (blue circles, orange squares, and green triangles). The black dotted line indicates truthful bidding $b(v) = v$ (the dominant strategy in second-price auctions). The blue dashed line shows the Bayes–Nash equilibrium bidding strategy reported in Table 3. The solid orange line is the LOESS-smoothed bidding curve.

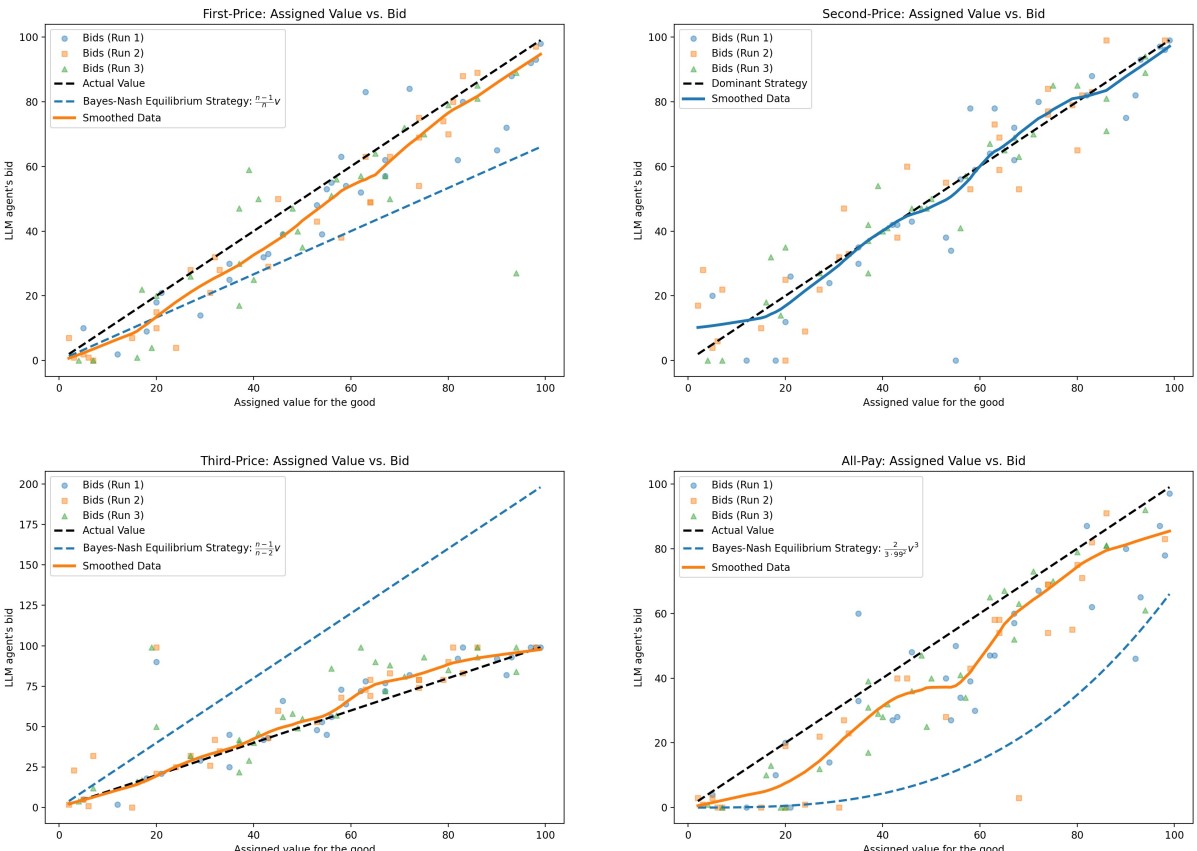

Figure 4: **Gemini2.5 bidding behavior across four auction formats under the IPV setting.** Bidders' private values are independently drawn from the uniform distribution on $\{0, \ldots, 99\}$, and Gemini2.5 bids are plotted against values (blue circles, orange squares, and green triangles). The black dotted line indicates truthful bidding $b(v) = v$ (the dominant strategy in second-price auctions). The blue dashed line shows the Bayes–Nash equilibrium bidding strategy reported in Table 3. The solid orange line is the LOESS-smoothed bidding curve.

